# Structure and Magnetic Properties of Fe-B-La-Al Alloy

**Sabina Lesz [1,\*]** , **Piotr Kwapuliński [2]** , **Małgorzata Karolus [2]** , **Klaudiusz Gołombek [3]** , **Bartłomiej Hrapkowicz [1]** , **Adam Zarychta [1]** , **Rafał Babilas [1]** , **Julia Popis [1]** and **Patrycja Janiak [1]**

1 Department of Engineering Materials and Biomaterials, Silesian University of Technology, Konarskiego 18a, 44-100 Gliwice, Poland; bartlomiej.hrapkowicz@polsl.pl (B.H.); adam.zarychta@polsl.pl (A.Z.); rafal.babilas@polsl.pl (R.B.); julia.popis713@onet.pl (J.P.); patrycja.janiak96@gmail.com (P.J.)

2 Institute of Materials Engineering, University of Silesia, ul. 75 Pułku Piechoty 1a, 41-500 Chorzów, Poland; piotr.kwapulinski@us.edu.pl (P.K.); malgorzata.karolus@us.edu.pl (M.K.)

3 Materials Research Laboratory, Silesian University of Technology, Konarskiego 18a, 44-100 Gliwice, Poland; klaudiusz.golombek@polsl.pl

\* Correspondence: sabina.lesz@polsl.pl

**Abstract:** Nanocrystalline magnetic materials are of great interest in order to meet the needs of electronics and electrical engineering. There are many possibilities to modify the synthesis parameters and chemical composition in order to obtain the most desirable magnetic properties and microstructure. The paper discusses an iron-based alloy with the addition of boron lanthanum and aluminium. The alloy was obtained by induction melting and casting with a melt-spinner. The main purpose of the work was to analyze the structure and properties of both the starting alloys in the form of ingots and the obtained tapes. X-ray diffraction (XRD), scanning electron microscopy (SEM), vibration magnetometry (VSM) and microhardness measurements using the Vickers method were carried out.

**Keywords:** magnetocaloric effect; nanocrystalline materials; melt-spinning; magnetic properties

## 1. Introduction

In the 21st century, when concerning drastically advancing climate change is a top priority, many countries have taken radical action to reduce the negative environmental impact of industry. Many of its most important branches-energy is constantly looking for new, promising solutions using sustainable ecological technologies [1,2]. Synthetic gaseous refrigerants have been applied to existing refrigeration equipment which has contributed significantly to promoting the degradation of the ozone layer [3,4]. The issue of magnetic cooling (MR), which can solve the problem of environmental pollution, is of increasing interest. This technique is based on the use of magnetocaloloric effect (MCE) during which isothermic magnetization of the material in a solid state occurs. This is followed by adiabatic demagnetization as a result of the removal of the magnetic field. Ultimately, this leads to a lower magnetic temperature. The process can be likened to isothermal compression and isotropic gas expansion [3–6]. This is implemented by the fact that the use of solid-based refrigeration techniques at room temperature, contributes to a significant improvement in the efficiency of cooling systems. This avoids the use of environmentally harmful gases in classical compression techniques [7,8].

There are many scientific reports suggesting that the best materials used in magnetic cooling are rare earth elements. Lanthanum and its alloys are of increasing interest to researchers. In relation to other representatives of rare earth elements, it is relatively inexpensive. Ferromagnetic materials lose their magnetic properties above the Curie temperature. This is extremely important because it is at this temperature that the most significant change in the temperature of the magnetism occurs. The Curie temperature for many rare earth elements is well below ambient temperature. In the case of alloys containing lanthanum the temperature can be adjusted to some extent which is crucial for an efficient cooling process, especially when a large temperature change is expected [7–9].

According to Pecharsky and Gschneider (2007), a single low Curie temperature can hider the possibility of the entire system temperature to rise above 25 °C [10].

Until now, the use of conventional polycrystalline ferromagnets and their alloys have been promoted. Amorphous iron-based alloys as metallic glasses have been the subject of interest research [11–14]. Many scientific papers [4,7,9–13] confirm that these alloys have good magnetic properties, as well as the use of iron beeing economically viable–it is inexpensive and easily accessible. According to Tang and others [9], in the context of magnetic cooling, amorphous alloys far outweigh their crystalline counterparts. The amorphous structure also allows the adjustment of the Curie temperature by directly manoeuvring the alloy composition [10]. In addition, the norrow hysteresis loop of this material proves the soft magnetic nature of these alloys which also implements low energy losses. They also have a low coercive force that is usually less than 100 A/m. Disorder of the structure determines better mechanical properties and corrosive resistance [4–15]. Boron [15] is an important addition that increases the plasticity of metallic glass and increases the glass forming ability. There are many publications on iron alloys with boron [16–20]. The oldest date back to the beginning of the last century and concern alloys with a crystalline structure. Already in 1926, promising properties of iron alloys with boron were noticed [21].

Luborsky's pioneering research [22] focused on the crystallization kinetics of the Fe-based amorpous alloy.

Metallic glasses are also a precursor to obtaining nanocrystalline materials. Many years of research on the design of an alloy with an optimal chemical composition has led to the obtaining of partially crystallized metallic glass Fe-Si-B with the addition of Cu and Nb with the lowest core losses [23].

This provides the basis for extending research towards the magnetic properties of lanthanum-containing alloys.

However, there are only some reports about the influence of Al and La on the Fe-based alloys [24–28]. Attention has been mainly focused on the study of multicomponent alloys such as bulk glassy $Fe_{73}Al_5Ga_2P_{11}C_5B_4$ alloys [24–26]

Jiang et al. [27] described the effect of the addition of lanthanum on the alloy containing iron and aluminum. Studies have shown that small addition of lanthanum already leads to visible fragmentation of alpha-Al grains. In addition, the significant positive effects of lanthanum on strength and electrical properties have been proven. These results obtained provided the basis for extending the research towards the magnetic properties of Fe-based alloys with a small amount of Al and La addition.

This work describes the basic magnetic properties of Fe-B-La-Al tapes produced by the melt-spinning method. The main topic of this work is to characterize the structure of the Fe-B-La-Al alloys in two various forms i.e., ingot and melt-spun tape. The structure of the ingot determines the magnetic properties of the tapes. Previous publications [14] have typically described alloys containing lanthanum, iron, silicon, and hydrogen, however, attention has been drawn to technical difficulties in their manufacturing and costs. There are few sources available on amorphous alloys, similar to those analyzed in the experiment. This makes the conducted research somewhat pioneering and may significantly contribute to expanding the current knowledge in the field of magnetic properties. After conducting extensive literature analysis, based on the sources mentioned before and among others (e.g., [9,24–30]), only sparse studies include the influence and effect of the base materials on the properties of the final materials obtained, such as the amorphous tapes [31].

One of the most essential aspects of this publication is the analysis of the base material structure.It has a very substantial effect on the properties of the final material, which is being obtained by a different process altogether (melt spinning).

## 2. Materials and Methods

In the study, it was decided to investigate the effect of the amount of lanthanum (0.1; 0.2 and 1 at.%) and small amounts of aluminum (0.5 and 1 at.%) on the structure

and properties of alloys obtained by the melt spinning method. The Fe-B-La-Al master alloys were prepared using Fe with a purity of 99.99% (MaTecK, Jülich, Germany), B with a purity of 99.9% (MaTecK, Jülich, Germany) La, and Al with a purity of 99.9% (Alfa Aesar, Haverhill, MA, USA). The master alloy composition is shown in Table 1. A total mass of 20 g was melted in an induction furnace under an argon (Ar) atmosphere, with the purity of 99.999%, using an $Al_2O_3$ crucible inserted inside of a graphite susceptor. Due to the different melting temperatures of elements, the pre-ingots of Fe-B were prepared first, then the elements with the lowest melting temperature (La and Al) were added to achieve the final ingot (Fe-B-La-Al) for each composition. The starting materials were melted in an induction furnace, the alloys were refined until they were homogeneous. Subsequently, each of the molten alloys was poured into a flat ingot mold, ensuring that the obtained ingot was able to be easily crushed into pieces that would fit the dimensions of the casting crucible. The starting alloys prepared in this way were re-melted and cast in the form of tapes by the Bühler Melt Spinner SC apparatus (Edmund Bühler GmbH, Hechingen, Germany) at a wheel speed of 30 m/s [32].

**Table 1.** The chemical composition and designations of the tested Fe-B-La-Al alloy, at.%. The accuracy of the main elements is about 4 wt.% and minor and trace elements are in a range of 20–50 wt.%.

| | Elements, at.% | | | | |
|:---:|:---:|:---:|:---:|:---:|:---:|
| **Alloy Indications** | **Fe** | **B** | **La** | **Al** | **Other** |
| La_1 | 87 | 12 | 0.1 | 0.5 | 0.4 |
| La_2 | 87 | 12 | 0.2 | 0.5 | 0.3 |
| La_3 | 85 | 12 | 1.0 | 1.0 | 1.0 |

The chemical composition of the samples of the Fe-B-La-Al alloy given in Table 1 was determined by Energy Dispersive X-ray Spectroscopy (EDS). Trident XM4 EDS with 20 kV of accelerating voltage was used in conjunction with SEM Zeiss SUPRA 35 (Carl Zeiss, Jena, Germany). These alloys contain negligible amounts of other elements such as C and Si which are from samples preparation.

An induction furnace and a melt spinner were used to produce the samples, which through the process of rapid cooling, makes it possible to produce metastable alloys, e.g., nanocrystalline. Several research methods were used to analyze the structure and properties of tapes such as X-ray diffraction (XRD), scanning electron microscopy (SEM), vibration magnetometry (VSM), and microhardness measurements.

The X-ray diffraction (XRD) measurements were performed by using the PANalytical Empyrean Diffractometer with Cu-K$\alpha$ radiation ($\lambda$ K$\alpha_1$ = 1.5418 Å) and a PIXcell detector. The phase analysis was conducted using the HighScore Plus PANalytical software (Almelo, The Netherlands) integrated with the ICDD crystallographic database PDF4+ 2018 (International Centre for Diffraction Data, Newtown Square, PA, USA). Structure analysis and crystallite size determination of synthesized phase were performed by Rietveld analyses [33,34] (High Score Plus software [35]), and the Williamson Hall theory [36].

Zeiss 35 scanning electron microscope (SEM) with a voltage of 20 kV was used to determine the microstructure of ingots and morphology of tapes.

Magnetic hysteresis loops of the Fe-B-La-Al tape samples were determined using VSM Quantum Design PPMS-7 (Quantum Design, San Diego, CA, USA) at temperatures: 10, 40, 100, 200, and 300 K.

The microhardness of the powder samples was analyzed on a Future-Tech FM700 Vickers hardness tester (Kawasaki, Japan). The microhardness measurement of ingots and tapes was carried out using a load of 50 gf (HV 0.05) with a 15 s dwell time, making 5 measurements on each sample. Both the starting alloys in the form of ingots and tapes were subjected to X-ray diffraction method and hardness tests.

### 3. Results and Discussion

#### 3.1. Ingots

The quantitative and qualitative phase analysis based on the ICDD PDF4+ 2018 database and the structural characterization of Fe-B-La-Al ingots, have indicated the presence of mainly two basic phases: $Fe_\alpha$ and $Fe_2B$ in all samples (Figure 1).

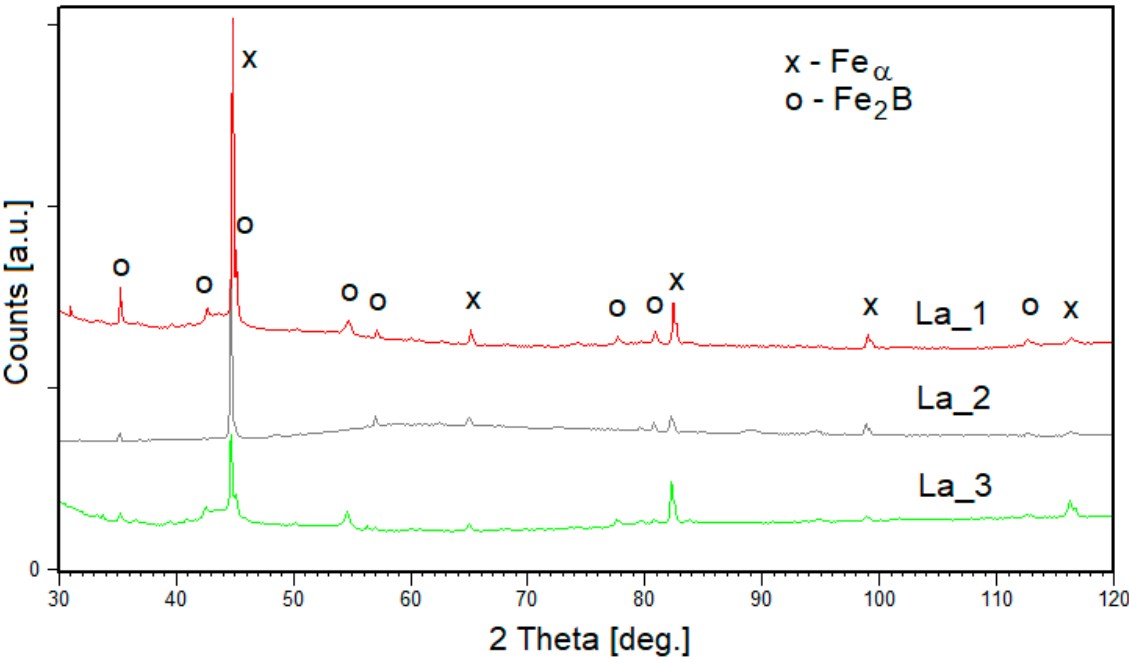

**Figure 1.** The X-ray diffraction patterns obtained for ingots of the Fe-B-La-Al marked with the La_1, La_2, and La_3, respectively.

The results of quantitative phase analysis were shown in Table 2. The La_2 sample containing almost only $Fe_\alpha$ phase (98 wt.%). Both samples La_1 and La_3 consist of about 70% of $Fe_\alpha$ phase and about 30% of $Fe_2B$ phase. The structural results were presented in Table 3. Generally, the starting materials are microcrystalline (crystallite sizes are greater than 1000 Å), only La_3 ingot has crystallites of the order of 300 Å. The structural analysis of the initial samples (ingots) (Table 3) shows that they are microcrystalline (crystallite size above 1000 Å) with slight deformation of the crystal lattice, as evidenced by very slight changes in the lattice constant (around 1%) and slight lattice distortions (of the order of 0.01–0.1%).

**Table 2.** Quantitative phase analyses of the main phases present in tested Fe-B-La-Al ingots (wt.%).

| Phase | $Fe_\alpha$ | $Fe_2B$ |
|---|---|---|
| Sample | | |
| La_1 | 72 | 28 |
| La_2 | 98 | 2 |
| La_3 | 71 | 29 |

**Table 3.** Crystallite size and changes of unit cell parameters of the main phases present in tested Fe-B-La-Al ingots.

| Sample | $Fe_\alpha$ | | | | $Fe_2B$ | | | |
|---|---|---|---|---|---|---|---|---|
| | Theoretical (ICDD PDF4+ Card: 04-016-6734) | Refined (RR) a/c [Å] | Crystallite Size D [Å] | Lattice Strain η [%] | Theoretical (ICDD PDF4+ Card: 04-003-2125) | Refined (RR) a/c [Å] | Crystallite Size D [Å] | Lattice Strain η [%] |
| La_1 | a = 2.8690 | 2.8688(5) | >1000 | 0.11 | a = 5.0990 c = 4.2400 | 5.1146(6) 4.2509(6) | >1000 | 0.08 |
| La_2 | Space Group: Im-3m | 2.8694(6) | >1000 | 0.02 | Space Group: I₄/mcm | 5.1018(2) 4.2488(3) | >1000 | 0.01 |
| La_3 | Crystallographic System: Cubic | 2.8690(8) | 303 | 0.11 | Crystallographic System: Tetragonal | 5.1041(4) 4.2491(1) | ~1000 | 0.03 |

The SEM microstructure of the La_1, La_2, and La_3 ingot was presented in Figure 2a–c. Two phases can be distinguished in the examined alloys: αFe marked as I and eutectic phase αFe + Fe$_2$B marked as II (Figure 2a–c).

(a) Phase I                                                                 Phase II

(b) Phase I                                                                 Phase II

(c) Phase I                                                                 Phase II

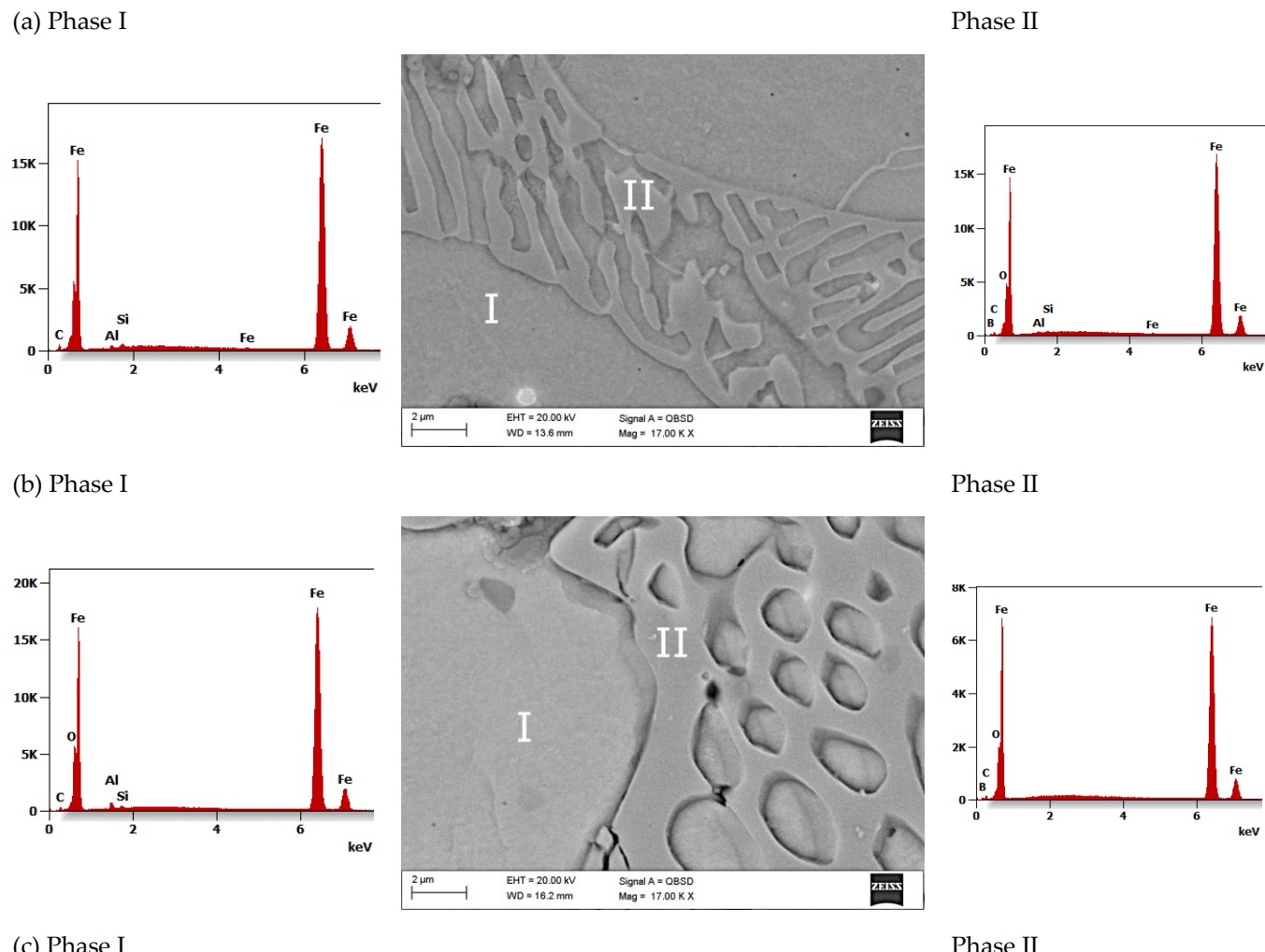

**Figure 2.** *Cont.*

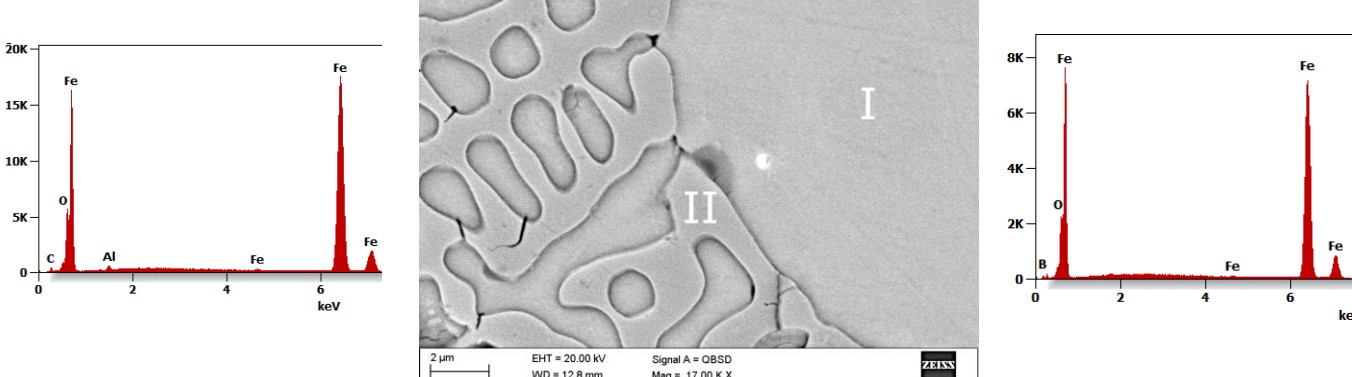

**Figure 2.** SEM image of the ingot structure in the center and the chemical composition of phase I and II La_1 (**a**), La_2 (**b**), La_3 (**c**).

The average values of phase I and phase II microhardness measurements of ingots together with the calculated errors are shown in Table 4.

**Table 4.** Average values and errors of microhardness of the Fe-B-La-Al alloy ingots.

| Sample | La_1 | La_2 | La_3 |
|---|---|---|---|
| Microhardness of phase I [HV] | $276 \pm 96$ | $256 \pm 72$ | $201 \pm 19$ |
| Microhardness of phase II [HV] | $365 \pm 40$ | $357 \pm 41$ | $339 \pm 23$ |

The obtained ingot microhardness results indicate that phase II has higher hardness than phase I in each of the alloys.

The microhardness value of phase I (solid solution based on the $Fe_\alpha$) varies between 201 and 276 HV. Phase II (eutectic mixture of $Fe_\alpha$ and $Fe_2B$ phase) has the highest hardness in the La_1 alloy and the value is 365 HV. With an increase of aluminum concentration-in the La_3 sample—(instead of 0.5 it will be 1 at.%), The hardness of phase I decreases. However, increasing the amount of lanthanum (0.1 -> 0.2 -> 1 at.%) causes a gradual decrease in hardness in phase II (Table 5).

**Table 5.** Quantitative phase analyses of the main phases present in tested Fe-B-La-Al alloy tapes (wt.%).

| Phase Sample | $Fe_\alpha$ | $Fe_2B$ |
|---|---|---|
| La_1 | 78 | 22 |
| La_2 | 81 | 19 |
| La_3 | 96 | 4 |

*3.2. Tapes*

The X-ray diffraction (XRD) patterns obtained for the tapes of the La_1, La_2, and La_3 are presented in Figure 3. The phase analysis (ICDD PDF4+ 2018) has indicated the presence of two phases: a solid solution based on the $Fe_\alpha$ exclusively in the body-centered cubic (bcc) phase and $Fe_2B$. The results of quantitative phase analysis were shown in Table 6. It can be seen that the quantitative proportion of the phases depends on the degree of crystallinity of the products.

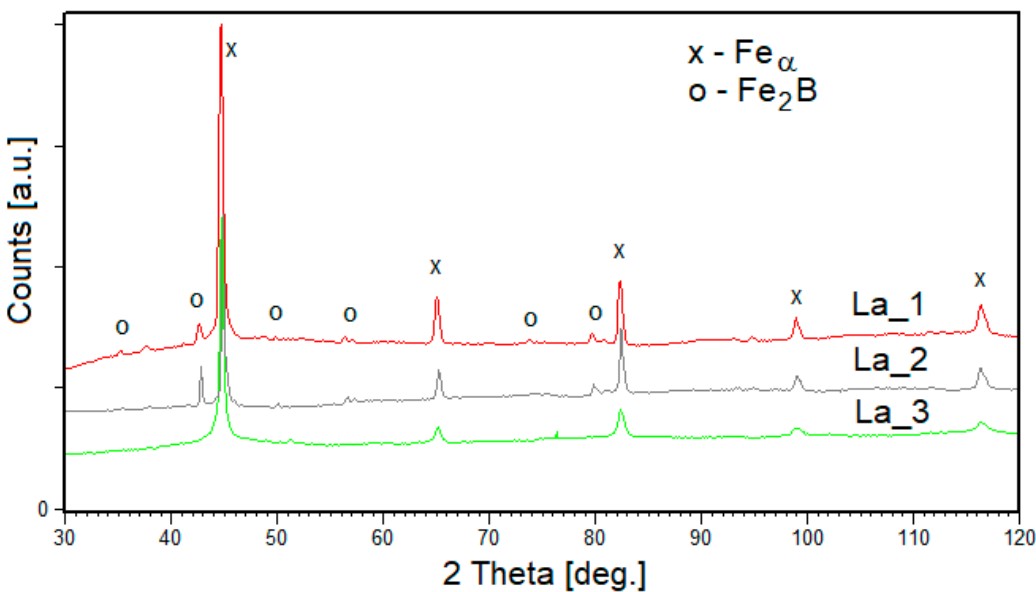

**Figure 3.** The X-ray diffraction patterns obtained for tapes of Fe-B-La-Al alloys marked with the La_1, La_2, and La_3, respectively.

**Table 6.** Crystallite size and changes of unit cell parameters of the main phases present in tested Fe-B-La-Al alloy tapes.

| Sample | Fe$_\alpha$ | | | | Fe$_2$B | | | |
|---|---|---|---|---|---|---|---|---|
| | Theoretical (ICDD PDF4+ Card: 04-016-6734) | Refined (RR) a/c [Å] | Crystallite Size D [Å] | Lattice Strain η [%] | Theoretical (ICDD PDF4+ Card: 04-003-2125) | Refined (RR) a/c [Å] | Crystallite Size D [Å] | Lattice Strain η [%] |
| La_1 | a = 2.8690 | 2.8671(4) | 231 | 1.13 | a = 5.0990 c = 4.2400 | 5.1136(3) 4.2428(5) | 160 | 0.20 |
| La_2 | Space Group: Im-3m | 2.8709(1) | 416 | 0.06 | Space Group: I4/mcm | 5.1085(2) 4.2478(3) | 538 | 0.05 |
| La_3 | Crystallographic System: Cubic | 2.8703(4) | 200 | 0.16 | Crystallographic System: Tetragonal | 5.1388(2) 4.1991(4) | 160 | 0.20 |

In the case of the La_2 ingot and La_3 tape, the diffraction patterns show broadening of the diffraction lines characterizing the Fe-based solid solution and the absence of Fe$_2$B diffraction lines (Figures 1 and 3). Thus, it can be assumed that the shift and broadening of the lines (Tables 3 and 6) indicate greater solubility of the alloy components in the basic structure of the solution. In the case of tapes (Table 6), the formation of nanocrystalline phases with a crystallite size of 160–500 Å is observed, which is accompanied by greater deformation of the structure visible in relatively larger lattice strain (0.15–1.13%).

The structural results were presented in Table 6. The La_3 sample containing almost only a solid solution based on Fe$_\alpha$ (96 wt.%) is characterized by the presence of its smallest crystallites of the order of 200 Å. A similar size of crystallites is determined for the La_1 alloy, as well. The largest crystallites are formed for the La_2 sample and reach the size of more than 400 and 500 Å for the solid-state solution based on Fe$_\alpha$ and Fe$_2$B, respectively.

The production of tapes in the melt-spinning process facilitates the migration of the alloy components, as a consequence of which the components build into the crystal lattice of the solution (0.06–1.13%) and Fe$_2$B phases (0.05–0.2%). It can be noticed that the smallest distortions of the crystal lattice are observed for the largest crystallite sizes of the obtained phases (Table 6-Fe$\alpha$-based solution: D = 416Å -> η = 0.06% and Fe$_2$B D = 538Å -> η = 0.05%). This may prove the relaxation of the crystal structure accompanying the growth of grains.

A very slight variation of the lattice strain occurs in both phases present in the La_2 alloy. This may be related to the gradual growth of crystallites and the formation of a stable microstructure.

The SEM images of the fracture morphology of the La_1 (a), La_2 (b), and La_3 (c) alloy tapes and an example of the surface of the La_3 (d) are shown in Figure 4. Their thicknesses are marked on the breakthroughs of the tapes (Figure 4a–c). The tape La_2 had the greatest thickness of 32.72 µm (Figure 4b) and the tape La_1 of 19.70 µm the smallest (Figure 4a). The SEM fractography of the La_1, La_2, and La_3 alloy tapes showed chevron (Figure 4a), vein pattern morphology (Figure 4b), and prevailing smooth fracture (Figure 4c).

(a)

(b)

(c)

(d)

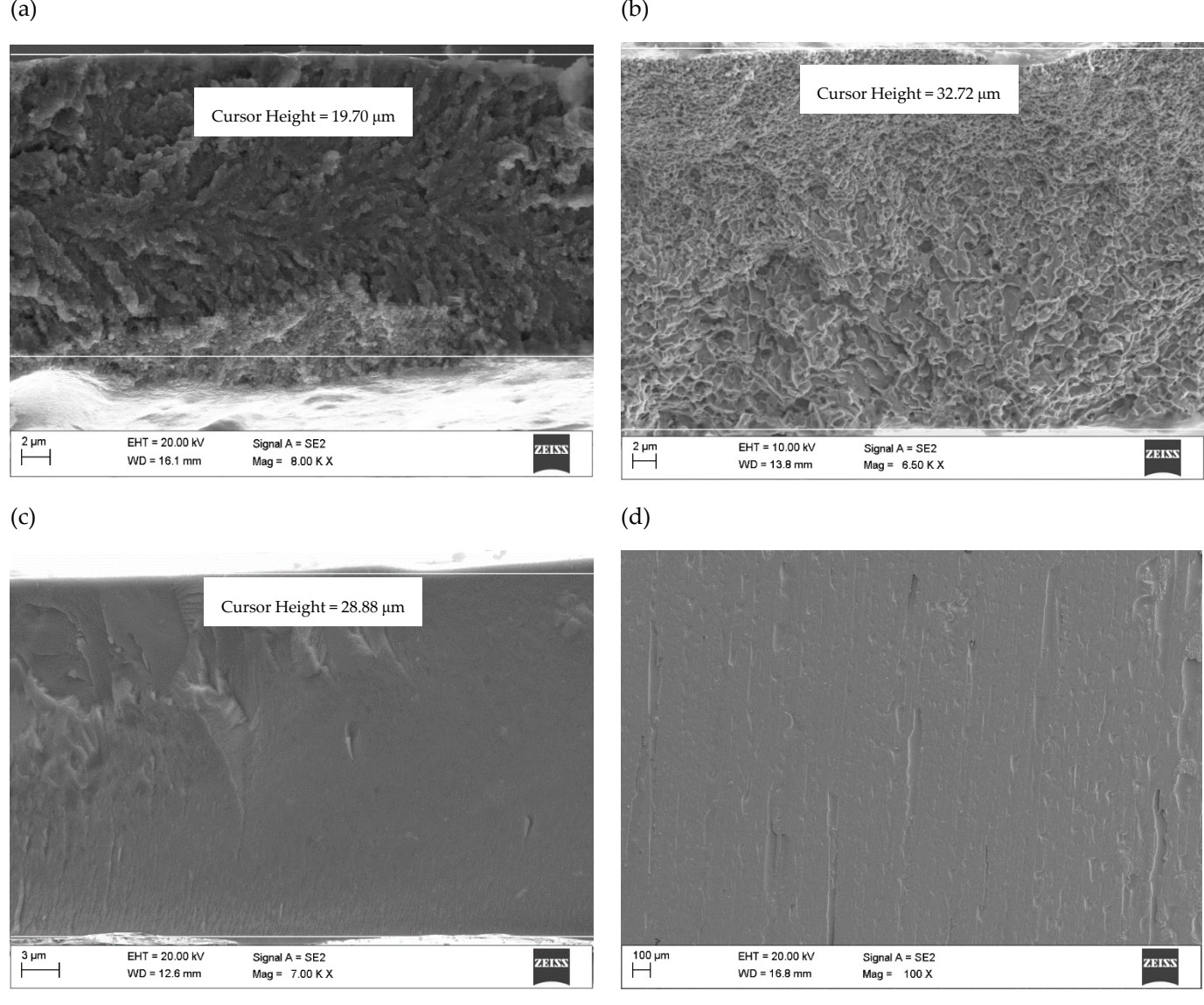

**Figure 4.** SEM image of the fracture morphology of the La_1 (**a**), La_2 (**b**), La_3 (**c**) alloy tapes with its thickness indicated. SEM image of the surface of the La_3 alloy tape (**d**).

The results of the study of magnetic properties (coercive force Hc and saturation magnetization Ms) of the alloy samples are presented in Table 7. They were determined on the basis of hysteresis loops. Chosen hysteresis loops of La_2 alloy samples are presented in Figure 5a,b. The asymmetric hysteresis loop, observed in Figure 5a,b is associated with the measurement methodology, it is the apparatus effect. However, in the multiphase systems, where different phases appear including the amorphous ones, the asymmetric hysteresis loop is a common occurrence, the multiple phases can be seen in Figure 3,

Tables 5 and 6 [37,38]. La_3 alloy tapes measured at various temperatures (from 10 to 300 K) decreases with increasing temperature. The smallest coercive force Hc for each alloy occurs at the temperature of 300 K (~ 27 °C), this value is 2.4 kA/m, 0.7 kA/m, and 0.2 kA/m for the La_1, La_2, and La_3 alloys, respectively (Figure 5a,b, Table 7). The highest value of saturation magnetization is for La_2 tapes. Tang et.al. [9] investigated the $Fe_{85}B_{12}La_3$ alloy. They have determined the magnetocaloric value of the alloy, as well as its coercivity field and magnetic saturation. It is worth mentioning that the magnetic saturation in their case has reached 255 $Am^2$/kg [9], whereas in our work we have obtained lower values, with a much lower amount of lanthanum (Tables 1 and 7). The magnetic properties of alloys are dependent on the type of crystal structure [39]. Without a doubt, the ferromagnetism of the alloy is not only dependent on the alloy's chemical composition but on atomic arrangement as well. The parameters of the alloy manufacturing and/or treatment e.g., cooling speed may influence the magnetic properties. The same mechanism can be observed in many alloys and metals [39].

**Table 7.** Magnetic properties (saturation magnetization-Ms and coercive force-Hc) of the La_1, La_2, and La_3 alloy tapes.

| Sample | Measurement Temperature [K] | Saturation Magnetization Ms [$Am^2$/kg] | Corecive Force Hc [kA/m] |
|---|---|---|---|
| La_1 | 10 | 178 | 4.3 |
| | 40 | 178 | 5.5 |
| | 100 | 181 | 3.6 |
| | 200 | 180 | 4.4 |
| | 300 | 177 | 2.4 |
| La_2 | 10 | 191 | 3.1 |
| | 40 | 191 | 1.9 |
| | 100 | 193 | 2.8 |
| | 200 | 191 | 1.8 |
| | 300 | 189 | 0.7 |
| La_3 | 10 | 142 | 1.0 |
| | 40 | 143 | 0.6 |
| | 300 | 136 | 0.2 |

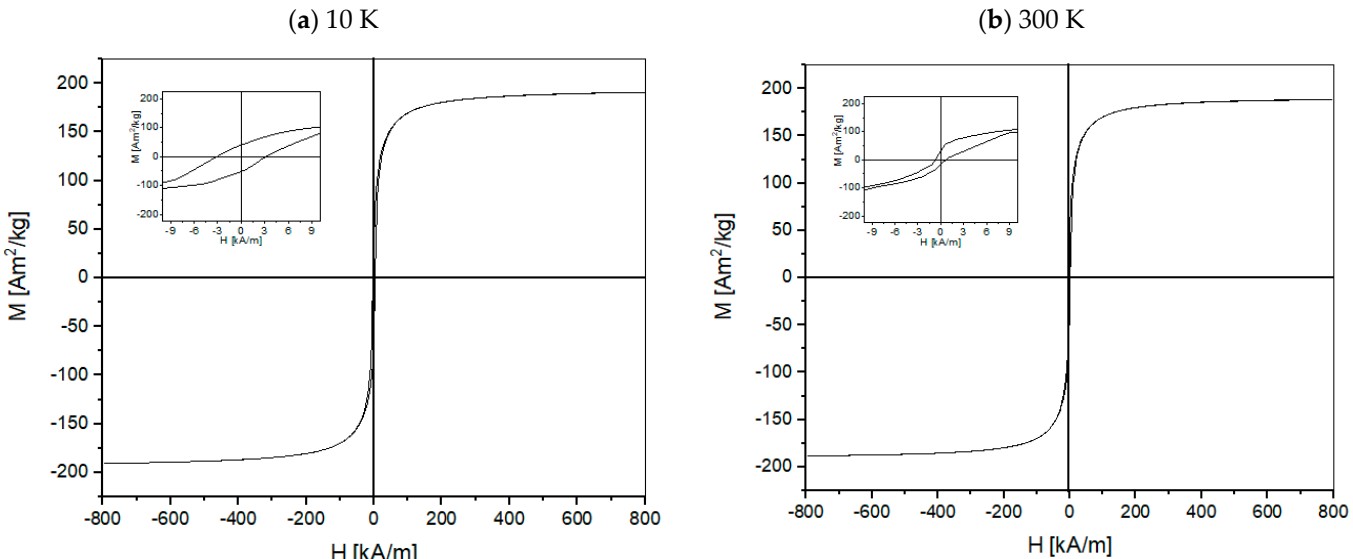

(**a**) 10 K    (**b**) 300 K

**Figure 5.** The hysteresis loops of La_2 alloy tape measured at various temperatures: (**a**) 10 K, (**b**) 300 K.

The magnetic properties are dependent of the phase composition) phase, crystallite size, strain, and the magnetostriction coefficient [29,40–46]. All of those values change

among the sample composition, as they are influenced by the varying content of the Fe-$\alpha$ and $Fe_2B$ phases. The phase quantity can be seen in Table 5. The Fe-$\alpha$ phase is magnetically soft, besides the presence of the $Fe_2B$ phase increases the intensity of the coercive force Hc. In Figure 6a,b. graphs showing the $Fe_2B$ content influence on the coercive force value Hc, and the relation between crystallite size D and the magnetic saturation Ms are presented. In the case of the tested alloys, no clear relationship was observed between the crystallite size D and the strain $\eta$ (Figure 6). Crystallites are embedded in an amorphous matrix which may result in the lack of a direct relationship between the size of the crystallites D and the strain $\eta$.

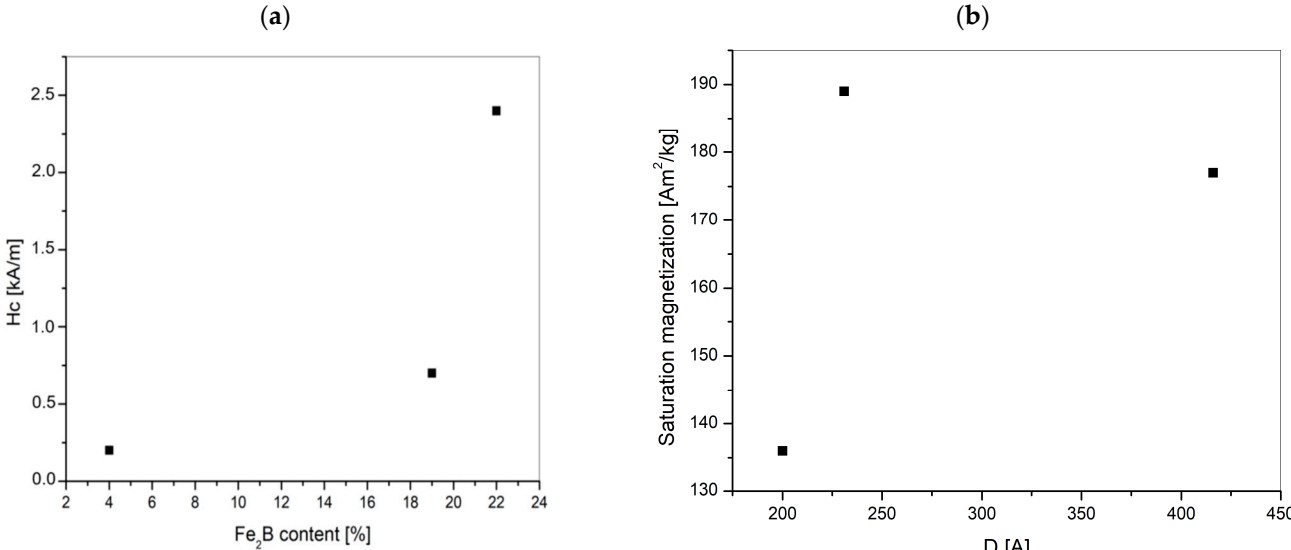

**Figure 6.** Graphs showing (**a**) the $Fe_2B$ content influence on the coercive force value Hc, (**b**) the relation between crystallite size D and the magnetic saturation Ms.

The results of measurements of microhardness of the Fe-B-La-Al alloy tapes are shown in Figure 6. Average values and errors are calculated and summarized in Table 8. Error bars (as depicted in Table 8) have been calculated by the statistics method, the standard deviation was designated.

**Table 8.** Average values of microhardness of the Fe-B-La-Al alloy tapes.

| Sample | La_1 | La_2 | La_3 |
|---|---|---|---|
| Average measurement [HV] | $530 \pm 56$ | $474 \pm 7$ | $1494 \pm 141$ |

It is found that the average microhardness value of the La_3 alloy tapes is the highest (1494 HV). The average microhardness of the La_1 alloy tapes is medium (530 HV) and the average microhardness value of the La_2 alloy tapes is the lowest (474 HV)-Figure 7, Table 8. The highest value of microhardness (1494 HV) was identified for the sample with the highest concentration of lanthanum (1 at.%) and aluminum (1 at.%).

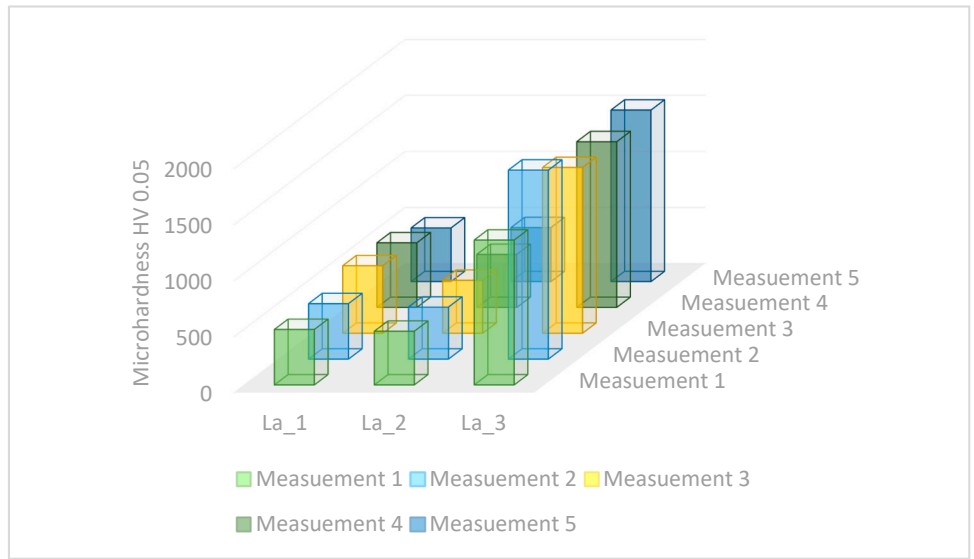

**Figure 7.** Results of measurements of the microhardness of tapes of the Fe-B-La-Al alloys.

**4. Conclusions**

- The X-ray diffraction analysis of the ingots and tapes of the Fe-B-La-Al alloys showed that the solid and $Fe_2B$ phases are present. In the structure of the samples, La_1 and La_3 ingots occur about 70% wt.% of $Fe\alpha$ phase and about 30% of $Fe_2B$ phase. The structure of La_2 ingot contains mainly $Fe\alpha$ phase (98 wt.%). The La_1 and La_2 samples in tape form consist of about 70–80 wt.% of $Fe\alpha$ phase and about 20–30 wt.% of $Fe_2B$ phase. The La_3 sample contains mainly $Fe\alpha$ phase.

- The structure of both La_1 and La_2 tapes consists of about 80% of solid solution based on the $Fe\alpha$ phase and about 20% of $Fe_2B$ phase. The structure of the La_3 sample containing mainly $Fe\alpha$ phase (96 wt.%). It can be seen that the quantitative proportion of the phases depends on the degree of crystallinity of the products. The largest crystallites are formed for the La_2 sample and reach the size of 400 and 500 Å for the solid solution $Fe\alpha$ type and $Fe_2B$, respectively.

- The SEM fracture morphology of the alloys annealed has a vein, chevron pattern, and smooth character.

- The lowest coercive force Hc occurs at the temperature of 300 K (~27 °C), this value for La_1 is 2.4 kA/m, for La_2: 0.7 kA/m and for La_3 - 0.2 kA/m. The highest value of saturation magnetization is for La_2 tapes. The magnetic properties of alloys are dependent on the type of crystal structure.

- The average microhardness value of the La_3, La_2, and La_1 alloy tapes is 1494 HV, 530 HV, and 474 HV, respectively. The average values of microhardness of the Fe-B-La-Al alloy ingots are significantly lower than tapes. The microhardness value of phase I (solid solution based on the $Fe\alpha$) varies between 201 and 276 HV. Phase II (eutectic mixture of $Fe\alpha$ and $Fe_2B$ phase) exhibits the highest hardness (365 HV) in the La_1 ingot.

- The highest value of microhardness (1494 HV) was identified for the sample with the highest concentration of lanthanum (1 at.%) and aluminum (1 at.%). The average microhardness of the La_1 alloy tapes is medium (530 HV) and the average microhardness value of the La_2 alloy tapes is the lowest (474 HV).

**Author Contributions:** Conceptualization, S.L.; Supervision, S.L.; Resources, S.L. and R.B.; Investigation S.L., P.K., M.K., K.G., B.H.; Formal analysis, P.K., M.K., K.G.; Validation, S.L. and B.H.; Writing—original draft, S.L., J.P., P.J.; Writing—review & editing, S.L., B.H., A.Z., R.B., J.P., P.J.; All authors have read and agreed to the published version of the manuscript.

**Funding:** The work was created as a result of the project of the Student Scientific Club of Magnetic and Composite Materials at the Faculty of Mechanical Engineering of the Silesian University of Technology, financed under the "Initiative of Excellence—Research University" program (1st competition).

**Conflicts of Interest:** The authors declare no conflict of interest.

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
