# Peer review of "Structure and Magnetic Properties of Fe-B-La-Al Alloy"

_magnetochemistry, doi:10.3390/magnetochemistry7090129_

Round 1

Reviewer 1 Report

  1. Page 2, line 54 please correct typing error “matrial”.
  2. By what method was the chemical composition of the Fe-B-La-Al alloy samples given in Table 1 determined?
  3. Page 7, line 186: please revised and correct the statement “The SEM images of the and fracture morphology”.
  4. The experimental details presented at page 8, line 193, were given in “Materials and Methods Avoid repeating these aspects presented above.
  5. Figures 5 can be summarize in one figure concerning the details of hysteresis curves for Hc with an inset of whole hysteresis as example.
  6. Figure 5 and Table 7 must indicate much more clearly whether it is the starting alloys or the rapidly cooled tapes!
  7. First conclusion is rather confused. Please rephrase!
  8. The correlation between the magnetic properties, on the one hand, and the structure and microstructure, on the other hand, is not sufficiently revealed.

Reviewer 2 Report

The present manuscript describes different characteristic properties of Fe-Ba-La-Al alloys in two different forms, say ingots and tapes. Many shortcomings can be found in this manuscript. In the next version of the manuscript, please address the following queries

  1. The title should be changed. It should be written based on composition or can be written with “La doping”
  2. What is the importance to choose two different form of the alloy (ingots and tape) to carry out a similar characteristic study? Kindly describe the motive of this study in detail.
  3. In line #82 Lanthanum spelling is wrong.
  4. Why La-2 sample shows different characteristics compared to the other two samples? The chemical composition of the samples changed in a systematic manner. Thy why is this anomaly?
  5. “The magnetic properties of alloys are dependent on the type of crystal structure [31].” Please clarify this statement in detail.
  6. What is the major significance of this work based on the experimental results? It is missing here. Add one table to do a comparison with other stoichiometry.
  7. Why in the La-2 sample, Fe2B peak is missing at the lower angle (in the ingots case)? Why in the La-3 sample, Fe2B peak is missing at a lower angle (in tape case)? Kindly explain these unusual characteristics.
  8. Please describe in detail the sample preparation methods of these six compositions.
  9. Is there any relation between crystallite sizes and lattice strain? Kindly explain your results
  10. What is the importance of the two phases here? Kindly explain it physically.
  11. Why magnetic hysteresis curves are not symmetric in higher temperatures? Are there any structural phase transitions? If yes please add references.
  12. How error bars have been calculated as depicted in Table 8?

Round 2

Reviewer 2 Report

Please submit the revised version of the manuscript in corrected format. Figures are tables are displaced here (Ex. Fig. 2, Fig. 5 and looks like some extra table format is added). It is very problematic to read this manuscript.

Author Response

According to your notes, all the unclear images have been reworked. Thank you.

Round 3

Reviewer 2 Report

The authors addressed my previous comments carefully and modified the manuscript accordingly. However, still few minor points are there in the manuscripts, which must be addressed before acceptance. The comments are summarized in the following:

  1. Please rephrase the lines and split it to short sentences. It is not clear. [Line #73 to #80]
  2. Split the sentence (line # 85)
  3. Please add the country origin of the companies MaTecK and Alfa Aesar.
  4. “In the case of the La_2 ingot…” this line is repeated twice (see line # 173 and #193). Delete one.
  5. How lattice strain is connected with microstructures? Please clarify in details or modify the like (line # 198)
  6. What is C La_3 alloy? (line #214)
  7. “The results of the study of magnetic properties (coercive force Hc and saturation magnetization Ms) of La_1, La_2 and 210 La_3 alloy samples determined on the basis of hysteresis loops are presented in Table 8.” Please split the sentences and write it in simpler form. (line #210)
  8. Which model has been employed to fit the Hc variation (see Fig. 6(a))? It is very difficult to understand the nature of the graph based only three points. It is better to show only scattered points only. The same is applicable for Fig. 6(b).
  9. Change the 3D graph format as depicted in Fig.7. In present form La-2 sample data points are completely shadowed by La-3 data points. Maybe it is better to show the data points in point format or by reducing the transparency of La-3 data points to make other data visible.
  10. “The magnetocaloric value of the alloy has been determined, as well as it’s coercivity field and magnetic saturation.” (line #217). Please delete the line or provide experimental results for the same.
  11. “All of those values change among the sample composition.” (line #225) Please explain it specifically.
  12. “However, in the multiphase systems, where different phases appear, including the amorphous ones, the asymmetric hysteresis loop is a common occurrence [37], [38].” (line # 213). It is very difficult to confirm a multi-phase system by measuring only M(H) loops. To clarify the statement either please add at least ZFC and FC M(T) graphs, or modify the sentence accordingly.
  13. Split this line “The quantitative…on all samples (Figure 1).” (line # 128)
  14. “lantan-containing alloys.” (line #66). Please correct it.
  15. Please correct the minor grammatical mistakes, and typos throughout the manuscript.

Author Response

Thank you for your consideration. Your comments have been included in the manuscript. For our responses, please see the attachment.
